# ‘Looking Back and Looking Forward’—Insights into the 20th European Doctoral Conference in Nursing Science (EDCNS)

**DOI:** 10.3390/nursrep15100350

**Published:** 2025-09-26

**Authors:** Lena Maria Lampersberger, Selvedina Osmancevic, Eva Pichler, Baptiste Lucien, Sebastian Rosendahl Huber

**Affiliations:** 1Institute of Nursing Science, Medical University of Graz, Graz 8010, Austria; selvedina.osmancevic@medunigraz.at (S.O.); eva.pichler@medunigraz.at (E.P.); sebastian.rosendahl-huber@fhgooe.ac.at (S.R.H.); 2Haute École Arc Santé, HES-SO University of Applied Sciences and Arts Western Switzerland, 2000 Neuchâtel, Switzerland; baptiste.lucien@he-arc.ch; 3Research and Development, University of Applied Sciences for Health Professions Upper Austria, Linz 4020, Austria

**Keywords:** nursing science, international conference, doctoral students, doctoral programme in nursing science

## Abstract

**Background**: The European Doctoral Conference in Nursing Science provides a unique platform for doctoral students in nursing and health sciences to present their research in a supportive environment. Celebrating its 20th anniversary, the 2024 conference embraced the motto “looking back and looking forward,” offering an opportunity to reflect on the development of nursing science and future challenges. **Results**: Held at the Medical University of Graz, Austria, the conference hosted 90 participants from 13 countries. It featured two keynote lectures, three workshops, 48 presentations, and a science slam. Abstract submissions underwent peer review to ensure the quality of presentations. The presentations highlighted key challenges and opportunities across nursing practice, healthcare work environments, education and digitalization in nursing, and health perspectives. Topics included, for example, workforce retention, artificial intelligence in nursing practice, leadership in error management, and culturally sensitive care. The keynotes emphasized the importance of patient and public involvement in research and the benefits of survey data in nursing science. Workshops imparted knowledge and skills regarding funding acquisition, guideline development, and effective research presentation. A science slam introduced innovative and creative ways to present research. **Conclusions**: The conference showcased the evolving landscape of nursing science, emphasizing the importance of evidence-based practice, supportive working conditions, and constructive collaboration. It demonstrated the enthusiasm and readiness of a new generation of researchers to advance nursing science in a rapidly changing healthcare environment.

## 1. Introduction

To begin with and to do justice to the motto of the described conference, we would like to look back at the development of nursing science and its value. In 1859, Florence Nightingale published her analysis of the mortality and morbidity of soldiers in the Crimean War, making her the first nurse scientist. The first doctoral programme for nurses was established at Columbia University, USA, in 1923, and slowly, the first funding opportunities for nursing research began to emerge. With the establishment of the American Nurses’ Foundation in the 1950s, the number of studies conducted in the field of nursing increased and the need for a nursing journal grew. Nursing Research was founded as the first nursing journal with a clinical rather than an educational focus [1]. In Europe, the development was not only different but slower and varied across the European countries. The end of the 1970s saw the first conferences in nursing research being held, and the Workgroup of European Nurse Researchers was founded. At the same time, academic nursing education and the foundation of doctoral programmes in Finland and Sweden marked the beginning of the “academization” of nursing education in Europe, i.e., transferring nursing education from secondary level training schools to tertiary academic institutions [2]. Because of these developments, we have a body of knowledge and evidence that informs nursing practice as well as nurses’ decisions and actions. This, in turn, prompted the development of evidence-based nursing, which has since improved the quality of patient care [1]. It is also worth mentioning that these celebrated historical developments also resulted in material changes for young researchers and doctoral students in the field of nursing, for instance, with regard to the growing academic field and the opportunity to present their research at academic conferences.

A conference for doctoral students in nursing science organized by doctoral students in nursing science—this is the distinct feature characterizing the European Doctoral Conference in Nursing Science (EDCNS). The EDCNS provides a unique forum for young researchers, and its 20th anniversary was held at the Medical University of Graz, Austria, on 20–21 September 2024. The two-day event welcomed 90 participants from 13 countries (i.e., Australia, Austria, Brazil, Finland, Germany, Indonesia, Ireland, Italy, the Netherlands, Spain, Sweden, Switzerland, United Kingdom). The conference is organized alternately in Austria, the Netherlands and Switzerland by students of the joint Doctoral Programme in Nursing Science (Medical University of Graz, Austria, Maastricht University, the Netherlands, and Bern University of Applied Sciences, Switzerland). This fosters a stress- and anxiety-free atmosphere and creates a safe space for young researchers to give scientific presentations to an international audience in the English language—often for the first time in their academic career.

The 20th anniversary of the EDCNS adopted the motto “looking back and looking forward”, which guided the subsequent proceedings. This special anniversary provided the opportunity to look back on over 20 years of success and celebrate the achievements of the past years, but also to obtain an insight into contemporary research. The conference was inaugurated over two decades ago by Professor Christa Lohrmann, now Head of the Institute of Nursing Science at the Medical University of Graz, Austria. When organizing the first EDCNS conference, Christa Lohrmann herself was a doctoral student at the Charité Berlin, Germany, and conceived and incorporated the idea of an international orientation, a supportive learning environment and its organization by doctoral students for doctoral students. One year later, the second conference was held at Maastricht University, the Netherlands, with a fellow student, Nynke de Jong, now Associate Professor at Maastricht University. These values have remained the core characteristics of this conference, and to the best of our knowledge, it is still a unique form of conference for nursing doctoral students in Europe [3]. The focus of this conference is on the students themselves, providing unique opportunities for its participants:

(1) Participants are welcomed in a supportive environment and can often take their first steps in giving scientific presentations to an international audience. (2) Support and feedback can be obtained from peers and professors to further develop presentation skills and knowledge. (3) Participants obtain knowledge relating to a wide range of topics through keynote presentations by renowned professors from around the world as well as in various workshops. (4) Due to the large number of doctoral students represented at the EDCNS, the participants are able to exchange ideas not only on certain topics but also relating to their experiences relating to their doctoral projects. Throughout the conference, participants are encouraged to build networks and friendships by providing time and space for such interactions throughout the event as well as an attractive social programme. (5) Furthermore, as numerous professors are attending as well, doctoral students have the opportunity to establish contact with them and to learn from their expertise.

It is also worth mentioning that this year’s EDCNS was held as a Green Event for the first time. This means that special attention was paid to environmentally friendly and resource-saving behaviour throughout the conference, both by the organization committee and the participants. In addition, the catering and formal dinner were provided by local businesses which focus on local products and have been certified as environmentally friendly businesses. As a result, the conference was awarded the Austrian Eco-label for Green Meetings & Green Events.

## 2. The Conference

The organizational team consisted of 5 doctoral students from Austria and Switzerland. The team members had various degrees of experience in organizing conferences. This way, less experienced students had the opportunity to learn from more experienced ones. The organizational team was also responsible for peer reviewing the abstracts with regard to meeting the quality criteria. A total of 55 abstracts were submitted for consideration. The review process was designed to ensure the quality and relevance of the research presented at the conference. As a result, 31 abstracts were selected for oral presentations, 17 for poster presentations, and three for science slam presentations. Additionally, two keynote speeches and three workshops were included in the programme. Table 1 provides an overview of all presentations held at the 20th EDCNS.

## 3. Keynotes

The keynote lectures were delivered by nursing scientists Birgit Heckemann from Gothenburg University, Sweden, and Silvia Bauer from the Medical University of Graz, Austria. Both keynote speakers are alumnae of the joint doctoral programme themselves. The presentation by [4] (Heckemann) focused on the growing necessity for patient and public involvement (PPI) in healthcare research. Despite its growing importance, PPI is not yet fully embedded in European healthcare systems and is being implemented with varying levels of commitment across the continent. Ref. [4] (Heckemann) elucidated the advantages and disadvantages of PPI, underscoring its capacity to facilitate empowerment for both researchers and stakeholders. She provided an overview of the importance, types, and phases of PPI, concluding with a discussion of practical resources for researchers to implement PPI in their projects [4]. The keynote by [4] (Bauer) addressed the role of survey data in nursing science, particularly the Nursing Quality Measurement survey, which was established in 1998. To date, the survey has yielded valuable insights into pivotal topics of concern in nursing, including pressure ulcers, incontinence, malnutrition, falls, physical restraints, and pain. Bauer demonstrated how this data has informed numerous doctoral research projects, resulting in notable advancements in nursing practice. She concluded that survey data not only facilitate nursing research but, as a result, also contribute to the improvement of nursing practice [4] (see Table 1).

## 4. Science Slam Presentation

The EDCNS 2024 was organized as a space to present research also in creative and engaging ways. While traditional scientific conferences often focus on formal presentations, the organizational team wanted to provide the opportunity to present research in a science slam, a popular format to creatively communicate research. This provides an opportunity to experiment with new formats, to step out of one’s comfort zone and, most importantly, to have fun while disseminating research. The result was a conference session full of energy, curiosity and new perspectives on nursing science, addressing often overlooked issues, challenging conventional thinking, or using compelling narratives to engage audiences. One presentation addressed the sensitive but crucial issue of sexual well-being in care, in her presentation “Why the sexual well-being of people with chronic illness is an important issue in nursing, and no, it’s not about the sexy nurse”. Ref. [5] (Igerc) explored how nurses can better support patients in this often-neglected aspect of care. Another presentation, “Cultural Competence: Illuminating Paths to Patient-Centered Care”, highlighted the importance of adapting nursing interventions to different cultural needs [5]. Ref. [5] (Gore) delivered a fascinating presentation on “Student Nurse to Super Nurse! Patient safety culture development in undergraduate nursing students”, showing how future nurses can develop a solid knowledge foundation regarding the topic of patient safety (see Table 1)

## 5. Oral and Poster Presentations

As shown in Table 1, the presentations were grouped into seven themes (i.e., healthcare as a workplace, holistic healthcare practices, nursing practice in focus, education and digitalization, different perspectives on health and health problems, multidisciplinarity and the care continuum, and caring for older people). The oral and poster presentations delivered by the doctoral students identified a plethora of challenges currently facing the nursing profession and, simultaneously, proposed potential solutions to these challenges. The overarching context of the presentations was the global demographic shift, characterized by an ageing population and an increased demand for nursing care, coupled with a decline in the proportion of the working population. This has engendered a need to improve the quality of care while simultaneously retaining the existing workforce [4,5,6,7].

Regarding the themes holistic healthcare practices, nursing practice in focus, and multidisciplinarity and the care continuum, presentations explored theories and definitions, advancements in nursing interventions and clinical decision-making highlighting the need for further research and standardized assessment tools [6,7]. For example, the definition of “usual care” varies considerably between contexts and individuals [6]. Furthermore, the characteristics of nurses’ autonomous supportive behaviour could not be explained by existing theories [6]. In their study, Ref. [6] (Bobbink et al.) found that nurse-led education can improve patient adherence to self-care routines, leading to better wound healing. The presentation by [6] (Van Dorst et al.) focused on home care needs assessments, identifying large variations in nurses’ assessments of home care needs, with a tendency to focus more on physical rather than psychosocial aspects. Studies on delirium [6], dysphagia [6], and cancer-related dyspnea [6] emphasized the importance of accurate assessment and intervention. The need for measurability is highlighted by the fact that more than 66% of hospital cases of delirium are not recognized as such. To counteract this, a study about the psychometric properties of the German version of the Ultra-Brief Confusion Assessment Method (UB-CAM) was presented [6]. Research on artificial intelligence in nursing assessments [6] and the development of culturally adapted tools, such as the Nursing Activities Score for Latvia, were also presented [7].

Regarding the theme healthcare as a workplace, presentations addressed the need to improve working conditions and to retain nurses, focusing on issues like presenteeism [6], workplace violence [6], and homecare workers’ health [6]. A review showed that almost half of nurses experience presenteeism, with influencing factors like workload, team culture, age, childcare, job insecurity, and leadership [6]. For the formal home care setting, over two thirds report back pain, and over one quarter report burnout significantly associated with work–life balance and verbal aggression [6]. Data shows that 42% of participants reported experiencing workplace violence, with 24% experiencing injuries, highlighting the significant consequences for formal caregivers [6]. Interventions to enhance job satisfaction, such as flexible shift planning and shared decision-making, were discussed [6] as well. A scoping review on nursing leadership and error management showed a strong link between positive leadership styles and reduced error rates, highlighting the need for leadership training [6]. In her study about barriers to healthcare workforce retention, Fiedler [6] found that rigid hierarchies, lack of appreciation, and poor working conditions negatively affect retention, with solutions including shared decision-making and improved flexibility. The role of advanced nursing practitioners in improving patient outcomes and supporting nursing autonomy was also highlighted [6].

Regarding education and digitalization, the integration of digital tools and artificial intelligence (AI) in nursing education and practice was a major theme in oral and poster presentations. In their study on AI for person-centred nursing quality, ref. [6] (Schönfelder et al.) showed that AI has a potential use in analyzing patient stories, though concerns were raised about maintaining the person-centred approach when using AI. According to another presentation, digital gaming in long-term care might positively impact physical and social functioning among elderly residents [6]. In their scoping review, ref. [7] (Alhonkoski et al.) identified the benefits of 3D technology, such as virtual reality and holograms, in improving student engagement, knowledge retention, and skill acquisition in anatomy teaching.

With regard to the theme perspectives on health and health problems, presentations examined the experiences of patients, family caregivers, and nurses, particularly in the context of cancer therapies and mental health. Some cancer therapies have severe adverse effects that warrant close monitoring. Even though much is known about the frequencies of the adverse effects of cancer therapies, there is still a dearth of knowledge regarding the experiences of patients, family caregivers, and nurses [6]. In their study on compassionate end-of-life care programmes, ref. [6] (Abraham et al.) found that end-of-life training can enhance staff competency, though emotional strain remains a challenge. To mitigate the possible burden on caregivers, a presentation showed that sense of purpose in life and a clear motivation for caring might alleviate this burden [6]. In contrast, mental health professionals might informally coerce patients into, e.g., taking medication, a topic about which not much is known to date [6]. In their study on neuropsychiatric symptoms in geriatric psychiatry, ref. [6] (Baumberger et al.) emphasized the need for non-pharmacological interventions to manage agitation and depressive symptoms in older adults.

## 6. Workshops

When beginning of a career in nursing science, reviewing the quality of studies, writing funding proposals and presenting research data are key competencies. Therefore, the three workshops offered at this conference focused on these topics. The first workshop by [5] (Hödl), an alumna of the joint doctoral programme, dealt with the topic of funding. It focused on enhancing the effectiveness of writing research proposals. The sessions covered critical aspects such as identifying motivations for success, the significance of collaborative efforts, formulating impactful research ideas, addressing imperfect grant calls, maintaining structured and motivated approaches, and celebrating successful submissions. Participants engaged in practical tasks to apply these principles, aiming to improve their proficiency in securing third-party funding for research projects [5].

The second workshop by [5] (Schoberer), also an alumna of the joint doctoral programme, discussed Guideline Development and GRADE (Grading of Recommendations, Assessment, Development and Evaluation). This workshop focused on the development of evidence-based clinical guidelines, specifically for fall prevention in Austrian hospitals and nursing homes. The process involved systematic literature searches, meta-analyses, and the application of the GRADE framework to assess the quality of evidence and to formulate recommendations. Participants were introduced to the basic steps of guideline development and the GRADE method, which is also useful for systematic reviews [5].

To emphasize the anniversary motto, ‘looking back and looking forward’, the organizational team took inspiration from the first EDCNS and endeavoured to bring it into the future by incorporating innovation and sustainability. At the first EDCNS, a workshop on Power Point presentations was held; a topic which still has relevance today, as numerous researchers struggle to design an easy-to-follow scientific presentation [8]. Therefore, a workshop titled “Present your research data—make it easy to follow!” was held by doctoral students. It emphasized the importance of effective data presentation in order to achieve research goals, including strategies for simplifying complex information to enhance clarity and impact. Participants learned techniques to make their data more accessible and persuasive to ensure their research findings are easily understood and compelling to their audience [5].

## 7. Conclusions

The presentations reflected the dynamic and evolving landscape of nursing science. The topics addressed during this conference are crucial for improving healthcare delivery, patient outcomes, and workforce sustainability [6,7]. Emerging themes such as digital health innovations, leadership development, and holistic care approaches highlight the ongoing transformation within nursing research and practice that started almost 200 years ago [1].

The 20th EDCNS highlighted current focal points of nursing research, which are also evident in international literature. For example, the urgent need to address demographic shifts towards an ageing population [9] and workforce challenges in nursing that come with it [10]. Furthermore, the conference underscored the importance of evidence-based practice, flexible working conditions, and shared decision-making to ensure high-quality care and to retain nursing professionals [11,12,13,14]. The insights and discussions from this conference provide a valuable foundation for future research and practice in nursing science. The conference also spotlighted a new, young, and motivated generation of nursing researchers who are ready to follow in the footsteps of established researchers. In the future, this conference and the doctoral students who attend it will continue to discuss current topics in nursing science, thereby contributing to the field.

Since the establishment of the EDCNS, more than 20 years ago, this format still constitutes a popular opportunity for doctoral students to gain experience in delivering presentations at academic conferences. By organizing the event in a way that is as ecologically sound and resource-efficient as possible, the conference is made fit for the future while taking into account the needs of our environment. This is also emphasized in international literature, which states that attending conferences as a doctoral student contributes to the student’s development as an independent scholar, for example. Furthermore, students can receive advice and support from their seniors, learn from their peers, and establish potential collaborations [15].

For the doctoral students in the organizational team of the EDCNS, the conference provides great opportunities to acquire organizational and project management skills. As first-time organizers of a conference, we learned how to secure funding, experienced the importance of sound marketing, and learned how to plan and execute a review process. These experiences equipped us with vital skills in academia and project management.

Furthermore, as the various members of the organizational team were in different stages of their doctoral studies, we had the possibility to learn from each other, as was also the case among the participants of the EDCNS. The conference was characterized by networking and providing mutual support and advice among peer doctoral students in different stages of their doctoral projects. This was also supported by the social programme of the conference. By providing opportunities to also meet in informal circumstances during the conference, networking, discussions, and helpful suggestions were made more easily achievable. Participants had the opportunity to find peers working on or with the same or similar topic or methods and to build lasting networks. The experiences from previous conferences have shown that participants stayed in contact even after many years, wrote applications for funding together and collaborated on topics of mutual interest in their further careers. In this conference, the formal and the informal programme items are therefore treated as equally important and receive the same careful planning and organization efforts.

For similar formats we recommend considering the situation of doctoral students and offering them opportunities for low-threshold networking, for example by organizing a forum for doctoral students within a conference or a separate lecture series for early-career researchers. This would allow young scholars to network, give presentations in a protected environment, and learn from each other.

## 8. Upcoming European Doctoral Conference in Nursing Science (EDCNS)

We are pleased to announce that the next European Doctoral Conference in Nursing Science (EDCNS) will be held in Maastricht, the Netherlands, on 17–19 September 2026. We are looking forward to continuing the tradition of fostering collaboration and innovation in nursing science. Updates on the next conference will be published under the following link: European Doctoral Conference in Nursing Science. José van Dorst can be contacted for further information regarding the 21st EDCNS (j.vandorst@maastrichtuniversity.nl).

## Figures and Tables

**Table 1 nursrep-15-00350-t001:** Overview of the conference presentations.

*Keynotes*					
[4] (Heckemann)			Unpacking a buzzword: Patient and Public Involvement in healthcare research		
[4] (Bauer)			Survey data in nursing science—a useful basis for numerous successful PhD-projects?		
** *Science Slams* **					
[5] (Igerc)			Why the sexual well-being of people with chronic illnesses is an important topic in nursing, and no, it’s not about the sexy nurse!		
[5] (Osmancevic et al.)			Cultural Competence: Illuminating Paths to Patient-Centered Care		
[5] (Gore)			Student Nurse to Super Nurse! Patient safety culture development in the undergraduate nursing student population		
** *Oral Presentations* **					
Health Care as a Workplace 1		Holistic Healthcare Practices 1		Nursing Practice in Focus 1	
[6] (Kepplinger et al.)	Understanding Employee Voice Behavior Through the Use of Digital Voice Channel in Long-Term Care: An Embedded Multiple-Case Study.	[6] (Apriyanti and Coyne)	Co-designing a culturally-informed Intervention to promote family-centred care in an Indonesian Paediatric Intensive Care Unit	[6] (Kero et al.)	A Systematic Review of Nursing Interventions for Dyspnoea Management among Inpatients with Cancer in Palliative Care
[6] (Fiedler)	Exploring Barriers and Opportunities for Effective Leadership in Addressing Healthcare Workforce Retention and Recruitment in Tyrol, Austria	[6] (Baumberger et al.)	The manifestation of neuropsychiatric symptoms in older psychiatric patients with cognitive impairment: An ethnographic case study	[6] (Bobbink et al.)	Making conscientious decisions to engage in venous leg ulcer self-management following nurse-led patient education
[6] (Miedema et al.)	Nurses’ preferences for modifiable hospital working conditions: a discrete choice experiment	[6] (Fink et al.)	Assessing Delirium in a Hospital in Germany—Psychometric Testing of the Ultra-Brief Confusion Assessment Method	[6] (Van Dorst et al.)	Practice Variation in hours, type and duration of home care in the assessments of one client case: a survey study
		[6] (Dentice et al.)	The concept of “Routine care” and “Usual care” in the nursing discipline: a multi-method study	[6] (Hutter)	Experiences of students in nursing professions with the end of life
Education and Digitalization 1		Nursing Practice in Focus 2		Different Perspectives on Health and Health Problems 1	
[6] (Schönfelder et al.)	PerCenAI—Development of Artificial Intelligence to analyse patient stories in the context of Person-Centred Nursing Quality	[6] (Steiner et al.)	Effective interventions for older adults (65+) transitioning from acute care setting to home: a systematic review	[6] (Abraham et al.)	What is the impact of the CEOL (Compassionate End of Life) program on end of life care from a staff, family and organizational perspective?
[6] (Schmied et al.)	Nursing students’ perception and communication of patient safety concerns—A multicenter cross-sectional survey study in Austria	[6] (Moreal et al.)	Safety and efficacy of subcutaneous administration of Beta-Lactams: A Systematic Review and Meta-analysis	[6] (Leinemann and Dunger)	Car-T Cell Therapy from the Perspective of Patients, Family, Caregivers and Nurses. A Multicentric Qualitative Study
[6] (Kukkohovi et al.)	The effectiveness of digital gaming on the functioning and activity of older people living in long-term care facilities: a systematic review and meta-analysis	[6] (Sterr et al.)	Advanced nursing practitioners’ impact on adult patients in acute care—partial results of a literature review	[6] (Kubitza)	Spirituality and care: a holistic perspective for family caregivers
		[6] (Botana Gronek et al.)	Navigating the Dimensions of Autonomy and Nurses Autonomy-Supportive Behavior Through the Lens of Nursing Theories.		
Healthcare as a Workplace 2		Holistic Healthcare Practices 2		Different Perspectives on Health and Health Problems 2	
[6] (Lucien et al.)	Workplace violence and their determinants toward formal caregivers in the homecare setting: A cross-sectional study.	[6] (Beeri et al.)	Exploring the definition and conceptualization of informal coercion in inpatient psychiatry: preliminary results of a scoping review	[6] (Batool et al.)	Ethical Contemplations: Patient and Public Involvement in Randomized Controlled Trials within Cancer Care Research Methods to involve people with dementia in health policy and guideline development: A scoping review.
[6] (Gerlach et al.)	Presentism among Nurses: An Integrative Review	[6] (Visintini et al.)	The experience of patients with acute graft-versus-host disease about oral medication adherence: a qualitative descriptive study	[6] (Admani et al.)	Incivility in the Therapeutic Radiography clinical setting in the UK: a concept analysis
[6] (Martins et al.)	Burnout and backpain in homecare workers and the association with psychosocial work environment—a national multicenter cross-sectional study	[6] (Zilezinski et al.)	Non-pharmacological interventions to prevent and treat delirium in critically ill children: a scoping review	[6] (Falkenstein et al.)	Felt-bodily communication as a medium for parents and their children with care needs: a neo-phenomenology perspective
		[6] (Palli et al.)	Development of clinical items to identify dysphagia in patients with dementia—A e-Delphi study		
** *P* ** ** *oster Presentations* **					
Education and Digitalization 2		Multidisciplinarity and the Care Continuum		Caring for Older People	
[7] (Alhonkoski et al.)	3D technologies to support teaching and learning in health care education—scoping review	[7] (Drought et al.)	What are midwives and sonographers’ understanding and experience of the informed consent process within antenatal screening for fetal anomalies.	[7] (Edede et al.)	A Systematic review of the Impact of care bundles on the incidence of pressure ulcer among at-risk older adults
[7] (Eronen et al.)	Can nurse students’ learning of infection prevention and control be promoted by applying principles of meaningful learning? A field experimental study	[7] (Waight et al.)	Construing compassionate nursing care—the perspective of nurses working in primary and community care.	[7] (Lahtinen et al.)	Nurses’ individualized care competence in older people’s nursing care—instrument development
[7] (Munoz)	Telemedicine Trust. Analyzing the Impact of Video Consultations on Healthcare Relationships: A Mixed-Methods Approach.	[7] (Penders et al.)	Measuring Autonomy Supporting Behavior: A Systematic Review	[7] Quartey	Professional action by nursing staff when carrying out toilet training to promote continence in geriatric care.
[7] (Qimeng et al.)	Women’s Experience and Needs on Using Digital Technologies for Gestational Diabetes Management: An Integrative Systematic Review on Patient Portal Features	[7] (Cerela-Boltunocva et al.)	Adaptation of the Nursing Activities Score in Latvia	[7] (Schmüdderich et al.)	Care problems and goals for improvement of a dementia-specific, nurse-led care model in German nursing homes—Results of a group Delphi study
		[7] (Bauernfeind)	Questionnaire development to assess the roles and responsibilities of nurses in ventilator weaning of infants in Austrian Pediatric Intensive Care Units (PICU)	[7] (Lazzarin et al.)	Antidepressant use, but not polypharmacy, is associated with worse outcomes after in-hospital cardiac arrest in older people
		[6] (Litvaitis)	Making Best Interest Decisions under Deprivation of Liberty Safeguards: A Q Methodological Study	[7] (Ballarin et al.)	The association of Frailty, Malnutrition, and mobility in 30-day mortality after hip fracture in older people
** *Workshops* **					
[5] (Hödl)			Convincing research proposals: experiences, tips, tricks, & go for it!		
[5] (Schoberer)			Guideline development and GRADE		
[5] (Pock and Lampersberger)			Present your research data—make it easy to follow!		

## Data Availability

No new data were created or analyzed in this study. Data sharing is not applicable to this article.

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
