# Peer review of "‘Looking Back and Looking Forward’—Insights into the 20th European Doctoral Conference in Nursing Science (EDCNS)"

_nursrep, 2025, doi:10.3390/nursrep15100350_

Round 1
Reviewer 1 Report
Comments and Suggestions for Authors
Dear authors,
thanks for submitting this conference report. There is few minor revisions you have to take into considerations.
Please do not use abbreviations in abstract.
In introduction I would suggest to start with a look back than start with actual conference, just inverse first two sections.
Author Response
Comment 1: Please do not use abbreviations in abstract
Response 1: Thank you for your comment. The abbreviations have now been removed.
Comment 2: In introduction I would suggest to start with a look back than start with actual conference, just inverse first two sections.
Response 2: Switching the order of the paragraphs in the introduction made it more concise. Thank you for this suggestion.
Reviewer 2 Report
Comments and Suggestions for Authors
The report for the conference “The 20th European Doctoral Conference in Nursing Science (EDCNS)” is well written. It offers a concise history of nursing science and the EDCNS, clearly outlines the conference objectives, and effectively summarizes the keynotes, presentations, workshops, and a science slam. Overall, it provides valuable insight into the conference.
Please note that in the Abstract section, the word “research” is incorrectly placed at the end of a new line following a large blank space. It should instead begin a new line properly, without unnecessary spacing.
Author Response
Comment 1: Please note that in the Abstract section, the word “research” is incorrectly placed at the end of a new line following a large blank space. It should instead begin a new line properly, without unnecessary spacing.
Response 1: Thank you for this comment. I compared the PDF version of the manuscript with the Word file, and it appears that there was an issue when converting to PDF. The Word file is correct and the PDF file should now also be correct.
Reviewer 3 Report
Comments and Suggestions for Authors
Thank you for giving me the chance to review this manuscript.
By connecting historical turning points to contemporary advancements, the introduction skillfully places the EDCNS within the larger context of nursing science. Also, a wide view of current nursing research trends is offered by the covering of a variety of topics, including workforce challenges, digitalization, holistic care, and patient-centered methods.
With this, I commend the author for coming up with this paper. However, there are some recommendations that I would like to point out to further enhance the paper.
- Long accounts of historical events and oral presentations are examples of sentences that are difficult to understand because they are lengthy and complex. I suggest summarizing broad trends and making links to global nursing science issues (such as workforce shortages and digital change) to go beyond descriptive reporting.
- The study is detailed, but it doesn't go into great detail about how the data might be interpreted.Examples of topics that could be critically examined in connection to future trends include leadership, AI in nursing, and workforce retention.
- Recommend to provide external scholarly sources to back up discussions regarding international nursing collaboration, doctorate education, and the influence of conferences on professional development.
- Suggest to extend the final section to more clearly consider how EDCNS will advance nursing science in the future and how lessons learned might be applied to similar projects around the world.
Author Response
Comment 1:
Long accounts of historical events and oral presentations are examples of sentences that are difficult to understand because they are lengthy and complex. I suggest summarizing broad trends and making links to global nursing science issues (such as workforce shortages and digital change) to go beyond descriptive reporting.
Response 1:
Thank you for your comment. We recognise that some of the sentences and paragraphs are rather long. We have shortened them and tried to be more concise.
Comment 2:
The study is detailed, but it doesn't go into great detail about how the data might be interpreted. Examples of topics that could be critically examined in connection to future trends include leadership, AI in nursing, and workforce retention.
Response 2:
Thank you for your comment. As this is a conference report, our aim was to provide an overview of the topics discussed at the conference. We did not intend to critically examine the speakers' results and findings; we have therefore cited the abstract book.
Comment 3:
Recommend to provide external scholarly sources to back up discussions regarding international nursing collaboration, doctorate education, and the influence of conferences on professional development.
Response 3:
We appreciate the reviewer’s suggestion. We have added international references to the beginning of the conclusion to emphasise the conference's contribution to current and pressing topics in nursing science. However, the discussion of what we learnt from the conference is based on our direct observations and experiences from the EDCNS project. We believe that including external sources here would not accurately reflect the unique context and insights gained from this project; therefore, no additional references have been added.
Comment 4:
Suggest to extend the final section to more clearly consider how EDCNS will advance nursing science in the future and how lessons learned might be applied to similar projects around the world.
Response 4:
Thank you for this tip. We have included more specific areas of application for our learning. Similar conference formats could address the needs of doctoral students by offering specialised sessions. By continuing the EDCNS, we will also contribute to nursing science by discussing current topics.